# A Critical Review of Radiation Therapy: From Particle Beam Therapy (Proton, Carbon, and BNCT) to Beyond

**DOI:** 10.3390/jpm11080825

**Published:** 2021-08-23

**Authors:** Yoshitaka Matsumoto, Nobuyoshi Fukumitsu, Hitoshi Ishikawa, Kei Nakai, Hideyuki Sakurai

**Affiliations:** 1Department of Radiation Oncology, Clinical Medicine, Faculty of Medicine, University of Tsukuba, Tsukuba 305-8575, Japan; knakai@pmrc.tsukuba.ac.jp (K.N.); hsakurai@pmrc.tsukuba.ac.jp (H.S.); 2Proton Medical Research Center, University of Tsukuba Hospital, Tsukuba 305-8576, Japan; 3Department of Radiation Oncology, Kobe Proton Center, Kobe 650-0047, Japan; fukumitsun@yahoo.co.jp; 4National Institute of Quantum and Radiological Science and Technology Hospital, Chiba 263-8555, Japan; ishikawa.hitoshi@qst.go.jp

**Keywords:** particle beam therapy, proton beam therapy, carbon-ion beam therapy, boron neutron capture therapy, combination therapy, drug delivery

## Abstract

In this paper, we discuss the role of particle therapy—a novel radiation therapy (RT) that has shown rapid progress and widespread use in recent years—in multidisciplinary treatment. Three types of particle therapies are currently used for cancer treatment: proton beam therapy (PBT), carbon-ion beam therapy (CIBT), and boron neutron capture therapy (BNCT). PBT and CIBT have been reported to have excellent therapeutic results owing to the physical characteristics of their Bragg peaks. Variable drug therapies, such as chemotherapy, hormone therapy, and immunotherapy, are combined in various treatment strategies, and treatment effects have been improved. BNCT has a high dose concentration for cancer in terms of nuclear reactions with boron. BNCT is a next-generation RT that can achieve cancer cell-selective therapeutic effects, and its effectiveness strongly depends on the selective ^10^B accumulation in cancer cells by concomitant boron preparation. Therefore, drug delivery research, including nanoparticles, is highly desirable. In this review, we introduce both clinical and basic aspects of particle beam therapy from the perspective of multidisciplinary treatment, which is expected to expand further in the future.

## 1. Background: Particle Beam Therapy as a Novel Radiotherapy

### 1.1. Promotion and Expansion of Particle Therapy Facilities

Particle beam therapy is a type of radiotherapy (RT). Particle beam therapy delivers a high radiation dose to tumors and as technology has improved, enables antitumor effects. The application of drug therapy is an integral part of development. We review the present status and future aspects of particle beam therapy in terms of drug therapy. Proton beam therapy (PBT) and carbon-ion beam therapy (CIBT) are common particle beam therapies. In 1946, Wilson suggested for the first time the use of accelerated protons in radiation therapy (RT) [1]. He investigated the depth-dose profile of protons accelerated at the cyclotron in Berkeley (CA, USA) and observed a steep increase in energy deposition at the end of the particle range, which is known as the Bragg peak. In 1958, the first clinical use of accelerated protons in the pituitary gland of 26 patients with advanced breast cancer was reported by the Lawrence Radiation Laboratory in Berkeley [2]. Several clinical studies have been conducted in the following decades. However, they were performed in a physics laboratory; thus, they had little beam time, and beam lines were not necessarily designed for medical use. The first medical treatment facilities to use PBT were constructed at the Clatterbridge Oncology Center (UK) in 1989, which uses a cyclotron and at Loma Linda University (CA, USA) in 1990, which uses a synchrotron. The first medical treatment facility for CIBT was initiated in HIMAC in Chiba (Japan), in 1994. As of 2019, over 250,000 patients have been treated with particle therapy (proton and carbon-ion [C-ion] beams). The number of treatment facilities has rapidly increased in recent years. Currently, treatment is provided at 100 facilities worldwide. The United States has the highest number with 40 facilities, followed by Japan with 24 facilities, Germany with seven facilities, and Russia with five facilities. There are 95 PBT facilities and 12 CIBT facilities [3]. Japan has more than 20 particle therapy facilities on its small land area and is one of the countries where particle therapy is becoming increasingly popular (Figure 1).

### 1.2. Physical Aspects of Particle Therapy

The dose distributions of protons and C-ions largely differ from those of photons, which are used in conventional RT. The biggest difference in physical characteristics is the existence of depth-dose distribution, the so-called Bragg peak, and by having this characteristic, particle beam therapy can give the maximum energy to the surroundings near the stop (cancer part) (Figure 2a). Although the dose distributions of protons and C-ions are very similar, the difference is that C-ion beams have narrower penumbra and longer fragmentation tails than proton beams [4].

In particle therapy, an extended Bragg peak (spread-out Bragg peak; SOBP) is formed to fit this Bragg peak to the size of the cancer (Figure 2b). Beam formation technology has evolved dramatically in recent years, and an increasing number of facilities are introducing an active scanning system (ASS) in addition to the conventional passive scattering system (PSS). PSS uses a rotating scatterer to expand the beam vertically and horizontally, and a collimator and bolus are used to optimize the beam to the cancer shape in front of the patient [5]. In contrast, the ASS uses a thin pencil beam, which is deflected vertically and horizontally by electromagnets, and the depth can be adjusted by changing the energy of the accelerator [6]. The advantage of the active scanning technique is its application in complexly shaped target volumes and reduction in the cost of manufacturing patient-specific devices, such as compensators, and the disadvantage is its larger lateral penumbra [7,8,9,10,11].

### 1.3. Biological Aspects of Particle Therapy

The concept of quality and quantity is important to understand the biological effects of PBT. The quality of PBT is defined as the amount of energy transferred per track (linear energy transfer; LET). Radiation with high LET can cause more severe damage to cells, represented by complex DNA damage that is difficult to repair. A parameter that quantitatively expresses the difference in biological effects due to this difference in LET is the relative biological effectiveness (RBE), which is defined as the dose ratio between the reference photon radiation and the particle radiation that produces the same biological effect. The international standard for proton RBE is 1.1; however, recent studies have shown that the distal end of a proton-SOBP shows a slightly higher RBE value depending on the increase in LET [12]. In contrast, carbon lines with high LET show high RBE values, with approximately 1.5 as the biological RBE and 3.0 as the clinical RBE [4,13,14]. Furthermore, it is important to recognize that although RBE is a useful parameter for expressing relative effects, its value may vary depending on the biological endpoint and the level of biological effect [15]. In addition, it is known that C-ions with higher LET radiation have a killing effect on hypoxic and generally radioresistant tumors [16]. The biological effect of radiation is enhanced in the presence of oxygen (oxygen effect) [17]. The oxygen enhancement ratio (OER) is a parameter that indicates the difference in biological effects depending on the presence or absence of oxygen. The OER is expressed as the ratio of the absorbed doses required to induce the same biological effect with and without oxygen. RBE and OER are LET-dependent variables (Figure 3), and the higher the LET, the higher the RBE, which reaches its maximum at approximately 100–200 keV/µm and then decreases (overkill effect). The OER shows a value of approximately 2.5–3.0 at low LET and asymptotically approaches 1 at high LET. High RBE [15,18,19,20,21] and low OER [16,22,23,24] values are characteristic biological properties of heavy-ion beams with high LETs.

## 2. The Role of Particle Beam Therapy in Multidisciplinary Treatments in Clinics

This chapter outlines particle beam therapy, which is often used in combination with drug therapy.

### 2.1. Esophageal Cancer

In 2020, it was estimated that 18,440 new patients developed esophageal cancer and 16,170 died in the United States [25]. Hence, the case fatality risk of esophageal cancer is still high, and the disease is currently the sixth leading cause of cancer-related deaths [26]. The gold standard treatment for localized esophageal cancer is surgery. However, some patients are unfit for surgery because of their advanced tumors (unresectable), poor general conditions (medically inoperable), refusal to undergo surgery, and concurrent chemoradiotherapy (CCRT) is considered for them as an alternative curative treatment to surgery [27]. However, cardiopulmonary toxicities after CCRT using photon beams are still associated with late adverse effects and would disturb the administration of high-dose chest irradiation in the treatments for esophageal and lung cancers [28,29]. Especially, a standard total irradiation dose in the treatment for esophageal cancer (50.4 Gy in 28 fractions) is less than that for lung cancer (60 Gy in 30 fractions) because larger irradiation fields are necessary to entirely cover the primary tumor and regional lymph nodes of thoracic esophageal cancers [30]. It is known that local and/or regional recurrences are the most frequent failure patterns after CCRT for esophageal cancer [31] and there are significant relationships between cardiopulmonary toxicity and irradiated volumes and doses in the lung and heart [32,33]. Therefore, dose escalation to the target while avoiding unnecessary doses to the lung and heart is a reasonable approach to improve treatment outcomes for esophageal cancer.

PBT offers advantageous physical properties to RT for the treatment of various cancers and can reduce the radiation dose and irradiated volume in organs at risk, such as the lung and heart, in RT for esophageal cancer compared with photon therapy [34,35]. We initially reported the usefulness of PBT alone or a PBT boost following photon RT without concurrent chemotherapy for patients with unresectable esophageal cancer [36]. In the case of PBT alone, the median total dose was 80 Gy RBE (Gy [RBE]), but patients were given a median dose of 46 Gy using photons with a boost to 80 Gy (RBE) using protons. Thereafter, CCRT using cisplatin/5 fluorouracil added to a median dose of 60 Gy (RBE) using PBT alone was performed in 40 patients, and no grade 3 cardiopulmonary toxicities were observed in our recent study [37]. In a Japanese multicenter retrospective study, clinical outcomes obtained from 202 patients (195 with squamous cell carcinoma and seven with adenocarcinoma) treated between 2009 and 2013 at four institutes were investigated. The 5-year rates of overall survival (OS) and local control of patients with stages I and II were 75.4% and 74.0%, respectively, and the corresponding rates of stages III and IV were 36.8% and 53.8%, respectively. Furthermore, grade 3 cardiopulmonary toxicities were observed in only three (1.5%) patients [38]. Table 1 shows that the incidence of cardiopulmonary toxicities in CCRT for esophageal cancer after PBT was less than that after photon RT [39,40,41,42,43]. A recent randomized study compared the total toxicity burden (TTB) and progression-free survival (PFS) between intensity-modulated radiotherapy (IMRT) and PBT and revealed that there was no significant difference in PFS between the two arms. However, the mean TTB, which was the other primary endpoint of the study, was 2.3 times higher for IMRT (39.9; 95% highest posterior density interval, 26.2–54.9) than for PBT (17.4; 10.5–25.0) and the PBT arm experienced numerically fewer cardiopulmonary toxicities and postoperative complications [43].

Recently, the impact of maintaining host immunity on patient survival has brought focus on CCRT for esophageal cancer, and lymphocyte count during CCRT may be a representative landmark for predicting survival [44,45]. Since PBT avoids unnecessary doses to the body of patients compared with photon RT, the studies showed that lymphocyte counts during PBT were well maintained and OS after PBT tended to be better than that after photon RT [45,46]. A pencil beam scanning method that provides a better dose distribution than passive scattering, which reduces the doses to the normal tissues close to the target, is currently available at most institutes, and future prospective studies may further confirm the true efficacy including survival outcomes of PBT for esophageal cancer.

### 2.2. Pancreatic Cancer

Pancreatic cancer is well known as one of the cancers with a poor prognosis and remains challenging to treat. According to a survey, there are an estimated 57,600 new patients and 47,050 deaths worldwide annually [26]. Surgery is the gold standard for the curative treatment of pancreatic cancer, and chemotherapy for pancreatic cancer has made great strides in the last few decades. The role of RT is to control local lesions, such as original tumors and locoregional lymph node metastases. Since the pancreas is surrounded by organs that are at risk, such as the stomach and duodenum, it is difficult to deliver a higher radiation dose for conventional photon RT. Therefore, particle beam therapy, which has an excellent dose concentration, is highly recommended for the treatment of pancreatic cancer and is mainly used for curative therapy of unresectable diseases and preoperative therapy for resectable diseases.

There have been few reports of particle beam therapy for pancreatic cancer before 2010, but it has been increasing rapidly since then. This is largely due to advances in chemotherapy and technological advances in particle beam therapy. Table 2 shows a review of concurrent PBT combined with chemotherapy for pancreatic cancer. Hong et al. conducted preoperative CCRT using proton beams and capecitabine in 25 patients with resectable pancreatic cancer and reported a 75% 1-year OS rate in 2011 [47]. After that, some clinical studies using proton beams commonly combined with capecitabine, gemcitabine, 5-FU, and S-1, have been reported, and Hong et al. reported a 2-year OS rate of 42% in their subsequent report in 2014 [48]. In the curative treatment for the unresectable or borderline resectable disease, the 1- and 2-year OS rates after PBT were 62–77.5% and 40–50.8%, respectively [49,50,51,52]. Maemura et al. performed induction chemotherapy followed by CCRT using proton beams. In their study, gemcitabine was used for induction therapy, and S-1 was concurrently combined. The irradiation dose was either a standard dose of 50 Gy or escalated dose of 67.5 Gy (RBE) with 25 fractions and achieved a 1-year OS rate of 80% [53]. Regarding toxicity, gastrointestinal ulcers have been reported as a late adverse event in some studies [48,49,53]. Pancreatic cancer using CIBT has been reported at several institutions in Japan, and Kawashiro et al. summarized the data as a retrospective multi-institutional study [54]. They investigated 72 patients whose irradiation dose was 52.8 and 55.2 Gy (RBE) with 12 fractions and concurrent chemotherapy was performed in 56 patients using gemcitabine or S-1. The OS rates were 73% and 46% at 1 and 2 years, respectively, and late adverse events (ulcers) were found in one patient. Using in vitro and in vivo studies, Sai et al. demonstrated that C-ion beam combined with gemcitabine had a superior potential to kill pancreatic cancer stem-like cells [55]. Chemotherapy for pancreatic cancer has improved prognosis with the advent of gemcitabine and has since been further improved with the combination of other drugs, such as nab-paclitaxel [56,57]. In addition, FOLFIRINOX has been demonstrated to have a higher antitumor effect than gemcitabine [58]. To date, the use of FOLFIRINOX in combination with particle beam therapy has been uncertain due to its strong side effects, however, in Europe, a prospective study on the combination of FOLFIRINOX as induction therapy for preoperative CIBT was conducted [59]. With regard to the technical progress of particle beam therapy, the concomitant boost [49,50,52,60], active scanning [61], and layer-stacking boost techniques [62] have been attempted.

### 2.3. Prostate Cancer

Prostate cancer (PC) screening and early diagnosis using prostate-specific antigen have indicated that there are approximately 192,000 new patients per year in the US [26]. In Japan, the annual number of newly diagnosed patients also increases every year, and over 12,500 PC-related deaths were estimated in 2019 [63]. Furthermore, almost half of Japanese patients with PC are over 75 years old, and RT plays an important role as a representative nonsurgical and curative treatment, especially for elderly patients. Since PC control after RT is dose-dependent [64], recent advances in modern RT techniques, including IMRT and brachytherapy, under image guidance can provide successful treatment outcomes by allowing delivery of higher doses to the prostate with less toxicity to the organs at risk, such as the rectum and bladder, compared with conventional three-dimensional conformal RT [65,66].

Protons and C-ions are charged particles and have been used for PC treatment through particle therapy for several decades. PBT was initially used as a tool for dose escalation after photon RT. The first randomized study compared the results between photon RT and PBT in patients with advanced PC. In the study, patients in the standard-dose and high-dose groups were treated with pelvic RT at 50.4 Gy/28 fr, followed by local photon RT at 16.8 Gy/8 fr (total dose: 67.2 Gy/36 fr) and local PBT at 25.2 Gy equivalents (GyRBE)/12 fr (total dose: 75.6 GyRBE/40 fr), respectively, and a better local control was achieved in the high-dose PBT group than in the standard-dose photon RT group (8-year local control rate: 73% vs. 59%) [67]. In addition, the incidence of grade 3 rectal bleeding was recorded in only 2.9% of the patients in the high-dose PBT group, although the corresponding rate of RTOG9413 using pelvic RT followed by local photon boost was 4.3% despite using a 70.2 Gy dose (5.4 Gy lower than the abovementioned 75.6 Gy PBT dose in the randomized study) [68].

The PROG95-09 trial tested a prostate PBT boost of either 19.8 GyRBE/11 fr (standard-dose group) or 28.8 GyRBE/16 fr (high-dose group) following 50.4 Gy/28 fr by local photon RT without regional irradiation for patients with stage T1–2 PC; the 10-year biochemical failures were observed in 32.4% and 16.7% of patients in the standard-dose and high-dose groups, respectively (*p* < 0.0001) [69]. Regarding toxicity, there were no significant differences between the standard-dose and high-dose groups; grade 3 gastrointestinal (GI) and genitourinary (GU) events were only noted in 1% and 2% of patients in the high-dose group, respectively, and 0% and 2% in the standard-dose group, respectively. At almost the same time, a randomized trial was conducted to determine the effectiveness of dose escalation using photon RT. The 10-year recurrence-free rates in the standard-dose group (70 Gy/35 fr) and in the high-dose group (78 Gy/39 fr) were 50% and 73%, respectively (*p* = 0.004); however, grade 3 GI and GU adverse events in the high-dose group were observed in 3.3% and 6.6% of patients, respectively, suggesting higher event rates compared with those in the 79.2 GyRBE dose of the high-dose group in the PROG95-09 trial described above [70] (Table 3).

In the same period, the efficacy of a combination of androgen deprivation therapy (ADT) with RT for intermediate- and high-risk patients with PC on not only disease control but also overall survival was shown in a randomized trial [71] and high-dose local PBT without photon RT yielded favorable outcomes regarding both PC control and adverse effects [72]. The clinical outcomes of local high-dose PBT combined with ADT were equal to or superior to those of local high-dose photon RT or PBT combined with pelvic or prostate RT. Furthermore, C-ions have been used for patients with PC as a local RT since 1995 at the National Institute of Quantum and Radiological Science and Technology Hospital, and the optimal RT dose and appropriate use of ADT in CIBT for PC have been investigated through several clinical trials [73]. At present, ADT is combined with CIBT for 2–6 months in intermediate-risk PC patients and for 2 years in high-risk PC patients based on previous phase I/II and II clinical trials; however, the appropriate indication criteria and duration of ADT use for PC in combination with high-dose RT, including PBT and CIBT, remains unknown.

Multi-institutional cohort studies involving four Japanese institutes were conducted to determine the appropriate use of ADT combined with PBT, and 520 intermediate-risk and 555 high-risk PC patients treated with PBT between 2008 and 2011 were analyzed [74]. Overall, the use of short-term ADT for ≤6 months improved the biochemical recurrence-free survival (bRFS), but no benefit of ADT for> 6 months was observed. The effectiveness of short-term ADT on bRFS was more evident in patients with two or three intermediate-risk factors than in those with a single factor. In contrast, short-term ADT for ≤6 months did not improve bRFS in the high-risk group. The study revealed that ADT for 12 and 21 months is preferable for patients with single and multiple risk factors, respectively, with high-dose PBT. Since ADT may cause dysfunction in lipid, glucose, and mineral metabolisms, especially in the elderly, prospective studies are necessary to determine the optimal ADT use and avoid unnecessary ADT administration in combination with high-dose RT for patients with PC.

### 2.4. Pediatric Cancer

The large difference between radiation therapy for aged patients and pediatric patients is the possibility of several complications. The following are typical complications and various complications that have been studied using photon RT: (1) Reproductive dysfunction: Sex cells are highly radiosensitive, and it is said that sperm cell depletion and morphological abnormalities can occur even at 10 cGy or less. Sanders et al. reported that 463 boys received total body irradiation of 10, 12, 14, and 15.5 Gy, and only five (1%) became fathers [75]. The Childhood Cancer Survivor Study (CCSS) examined 1915 patients after treatment for pediatric cancers, and the rate of miscarriage increased 3.6 times in cases with whole-brain and whole-spinal irradiation and 1.7 times in cases with pelvic irradiation [76]. (2) Cardiac complications: According to a CCSS, the relative risk of death from cardiac complications was as high as 11.9 (95% confidence interval (CI): 9.1–15. (3) in 2717 five-year-old or older survivors after chemotherapy for pediatric Hodgkin lymphoma. In cases where the mediastinum was irradiated with 42 Gy or more, it was reported to be 41.5 (95% CI: 18.1–82.1) [77]. PBT has a superior dose concentration with cytotoxicity equivalent to photons; thus, it is expected to be utilized for the treatment of patients with pediatric cancer. Moreover, although there are few clinical reports because secondary cancers develop years–decades later, PBT is expected to reduce secondary cancers. In Japan, PBT for patients with pediatric cancer became the first medical insurance coverage for all cancer diseases in 2016, which indicates high expectations for pediatric cancer treatment.

PBT is used for the treatment of cancers, such as brain tumors, neuroblastoma, and soft tissue tumors. PBT is not often administered alone and is often multidisciplinary with surgery or chemotherapy. Glioma, medulloblastoma, ependymoma, germinoma, and craniopharyngioma make up a lot of brain tumors. Eaton et al. compared the treatment results of photon RT and PBT for pediatric medulloblastoma. Both treatment protocols were the same as craniospinal and focal irradiation, with vincristine, cisplatin, cyclophosphamide, and lomustine. Second cancers were found in three of 43 patients in the photon RT group and none of the 45 in the PBT group [78]. They also investigated the endocrine result of photon RT and PBT in 77 patients and reported hypothyroidism as 69% and 23%, sex hormone deficiency as 19% and 3%, and any endocrine replacement therapy as 78% and 55%, respectively, and concluded that PBT may reduce the risk of some endocrine abnormalities [79]. Rhabdomyosarcoma (RMS) is a high-grade tumor characterized by local invasiveness and is the most common soft tissue sarcoma in pediatric patients. RMS can occur all over the body; however, the most common sites are the head and neck, followed by the pelvis. RMS is treated using a multidisciplinary approach. Clinical experiences in the treatment of pediatric RMS with PBT indicated safety and effectiveness with low acute toxicities and disease control comparable to photon irradiation [80,81,82]. Mizumoto et al. retrospectively summarized data from a multicenter study in Japan and reported the clinical outcome of 55 children aged 0–19 years (median, 5 years) who received PBT for RMS with doses ranging from 36–60 Gy (RBE). Surgical resection before PBT was performed in 41 patients (75%), and 53 patients (96%) received chemotherapy. The number of patients that enrolled during pre-PBT, pre- PBT and during PBT, and only during PBT was 17, 34, and two patients, respectively. The median follow-up time was 24.5 months. Acute toxicity of more than grade 3 was found in 16% of the patients, but they recovered well after PBT. A total of 87% of the patients experienced hematologic toxicities more than grade 3, which were very likely to be related to PBT [83].

## 3. Current Basic Research on Combination with Particle Therapy

### 3.1. Combination Therapy

In recent years, the content and quality of cancer treatment have been greatly developed along with advancements in science and technology. In other words, the introduction of new therapeutic concepts and methods into clinical practice through the invention and improvement in medical devices that enable high-precision RT, such as particle beam therapy, has brought progress in the cancer treatment. Early detection of early-stage cancers using modern diagnostic imaging technologies contributes to the improvement in survival rates of cancer patients; however, the clinical outcomes of many advanced-stage cancers using a single treatment method remain disappointing. Therefore, a multidisciplinary treatment combining RT with surgery, chemotherapy, molecular targeted therapy, or immunotherapy has been tested, especially for advanced-stage cancers, for a long period. How can a combined protocol be adapted for particle beam therapy? Recent basic biological studies have reported the possibility that higher sensitization effects can be demonstrated with proton beams, which have similar biological effects to photon beams. However, it is expected that combination experiments with C-ions, rather than protons, will show unexpected results. In this section, we summarize the latest findings on the combined effects of particle RT and various cancer treatment modalities.

### 3.2. Chemotherapy

Chemotherapy is often combined with RT for advanced-stage cancers, given before (neoadjuvant), at the same time (concomitant), and after (adjuvant) RT, and the efficacy and feasibility of the combination therapy have been confirmed in most of them. Many of the goals of RT and chemotherapy combinations are to treat locally and systemically distributed cancers, respectively.

However, it is well known that a combination of drugs and radiation can have local synergistic effects. As a typical example, concurrent chemoradiotherapy consisting of photon RT combined with platinum compounds for head and neck squamous cell carcinoma had a significant benefit of OS with a 10% reduction, but neoadjuvant or adjuvant chemotherapy did not [84]. In contrast, heavy particle beams with a high LET have a smaller sensitizing effect by various chemotherapeutic agents than photon beams. The sensitizing effect of camptothecin, cisplatin, gemcitabine, and paclitaxel, which are frequently used in the field of RT, is remarkable for X-rays, but extremely small for C-ions of 103 keV/µm [85]. Similarly, when glioblastoma (GBM) cells were treated with carbon wire and temozolomide (TMZ), only additive effects were observed [86]. In contrast, a remarkable synergistic effect with a decrease in the shoulder of the cell survival curve was observed in S-phase human colon cancer-derived cells when antimetabolite gemcitabine (2′,2′-difluoro-2′-deoxycytidine, dFdCyd) was combined with C-ions [87]. Proton beams, which exhibit biological effects similar to those of photon beams, can be expected to be more effective in combination with chemotherapeutic agents than C-ions, and concurrent chemoradiotherapy using protons is widespread in clinical practice. However, in recent years, differences between X-rays and proton beams have been reported, such as in gene expression and protein levels due to various double-helix DNA cleavages and DNA responses and disorders in the process of cell death [88,89]. It has been reported that overall DNA damage differs to some extent after proton and X-ray irradiation and that proton-irradiated cells preferentially undergo homologous recombination [90]. This result suggests that the distribution of DNA repair types differs between protons and X-rays. Our research also confirmed that the recovery phenomenon is less likely to occur after proton irradiation than after X-ray irradiation [91]. Furthermore, it has been reported that the sensitizing effect of CDDP may differ between X-rays and protons [92]. The combination index for CDDP in the three types of cells was 0.82–1.00 by X-ray, whereas it was 0.73–0.89 by proton beam, indicating that the proton beam has a stronger sensitizing effect. Analysis using the Cell Cycle Indicator System (fluorescent ubiquitination-based cell cycle indicator or FUCCI) also showed that CDDP and proton beam increased apoptotic cells and G2/M arrest induction more significantly than X-rays. This difference was particularly pronounced in radiation-resistant cells, suggesting that chemoradiotherapy with X-rays and protons may have different effects on radiation-resistant cancers.

### 3.3. Molecular Targeted Therapy

The toxicity of normal tissues is often increased by chemoradiotherapy, limiting the amount of drug or radiation that can be administered [93]. This fact is far from ideal, and more effective and less toxic combination therapies are needed for the systemic therapy combined with radiation. Against this background, research and development of molecular-targeted drugs are underway. It specifically inhibits cell survival, information exchange with surrounding tissues, and signal transduction associated with responses to internal and external stresses, including radiation. Recent developments in the fields of biotechnology and cell biology have revealed and rapidly elucidated signal transduction and intracellular crosstalk.

A suitable target molecule for a drug used for RT should be overexpressed in many tumors to be treated and not in the normal tissue surrounding the tumor [94,95]. Molecules involved in the mechanism of radioresistance are also ideal. Given these conditions, good examples of reasonable targets related to RT include epidermal growth factor receptor (EGFR), vascular endothelial growth factor (VEGF), phosphatidylinositol-3-kinase/protein kinase B/mammalian target of rapamycin (PI3K/Akt/mTOR) pathway, Ras pathway, and histone deacetylation. The efficacy of these molecular-targeted drugs in combination with photon therapy has been demonstrated in both basic research and clinically, for example, with EGFR inhibitors.

Although there is insufficient clinical data on the use of particle beams in combination with molecularly targeted drugs and the efficacy of these drugs is not yet clear, many interesting data have been reported from basic biological experiments using cells and mice. The number of relevant studies has increased from approximately 30 in 2010 to over 140 in 2020. The combination of the proton beam and a VEGF inhibitor, bevacizumab, decreased cell survival, increased apoptosis levels, and cell cycle arrest in melanoma cells [96]. The EGFR inhibitor gefitinib preferentially sensitized non-small cell lung cancer cells (H460 and H1299) to proton radiation compared to gamma radiation [97]. Cetuximab (Cmab) also preferentially sensitizes cancer stem cells (SQ-20B/CSCs) isolated from head and neck squamous cell carcinoma (SQ-20B) to carbon beam irradiation compared to photon irradiation [98]. Furthermore, C-ions + Cmab strongly inhibits the enhanced invasion ability of SQ-20B/CSCs, and it has been suggested that the combination therapy is a very promising treatment that suppresses the migration and infiltration process of all cancers as well as CSCs. A PARP inhibitor (PARPi), AZD2281, enhanced the effect of protons on human lung cancer cells (A549) and pancreatic cancer cells (MIAPaCa-2), inducing an increase in γ-H2AX counts and an increase in G2/M arrest [99]. Another PARPi, olaparib and niraparib, sensitized the cytotoxic effects of protons on lung, pancreatic, esophageal, and head and neck cancer cells as well as on X-rays [100,101,102]. These inhibitors have sustained DNA damage from irradiation, delayed apoptosis, prolonged cell cycle arrest, promoted aging, and had more synergistic effects on cells with high proliferation rates. In addition, PARP inhibitors have been effective against C-ions, and the PARPi ADZ2281 enhances the cell-killing effect of carbon rays on pancreatic cancer cells [103]. The combination of talazoparib and carbon beams dramatically reduced the number of GBM stem cells (GSCs), induced prominent and long-term G2/M block, and reduced GSC proliferation [104]. Owing to its multifunctional role and low expression in normal cells, Hsp90 inhibitors are considered a good target for cancer therapy; however, there are few reports on its effects in combination with particle therapy, and we found two reports on its combination with carbon beams [105,106]. 17AAG enhanced the cytotoxic effect on lung cancer cells and the antitumor effect on lung cancer transplanted tumors after C-ions to the same extent as X-rays [105]. This sensitizing effect was due to the inhibition of homologous recombination repair by 17AAG, and as a result, the strong induction of G2 arrest was confirmed. The combination of C-ions with different LET (14 and 50 keV/µm) and another Hsp90 inhibitor, PU-H71, showed higher or similar enhancement ratios compared to X-rays [106]. This result suggests a dependence of the enhancement ratio of molecular-targeted drugs on LET. Valproic acid, a histone deacetylase inhibitor (HDACi), sensitizes hepatocellular carcinoma cells to proton beams more strongly than photon beams by prolonging DNA damage and enhancing apoptosis [107]. In addition, Tsuboi et al. reported that another HDACi, suberoylanilide hydroxamic acid, enhanced the cell-killing effect of C-ions [108]. The combination of novel molecularly targeted drugs and particle radiotherapy has great potential to improve patient outcomes. However, the basic biological knowledge of this combination therapy is scarce, and further research is needed both in vitro and in vivo.

### 3.4. Nanoparticles

Metal nanoparticles (MNPs), such as gold and platinum, have been reported to exhibit radiosensitizing effects in photon-irradiated cells and animal models [109,110,111]. Experiments have shown that the mechanism is largely due to the interaction of photons with high-Z metal atoms to produce low-energy electrons (especially Auger electrons) [112,113], which in turn promotes the production of reactive oxygen species [114]. Currently, several attempts are being made to introduce nanoparticles into clinical practice.

However, because the interaction of charged particles is largely independent of the atomic number Z of the target, the sensitization of particle beams by MNPs is expected to be even smaller than that of photon beams. However, several experiments have confirmed the large sensitizing effect of the combination of proton beams with MNPs [114,115]. C-ions with high LET, in combination with magnetic nanoparticles, reduce the fluence per unit dose, making the probability of direct interaction at therapeutic doses unrealistic.

Furthermore, in recent years, the application of biomaterials, such as polymer nanofibers, for cancer treatment and drug delivery systems has increased [116,117]. Polymeric nanofibers are an exciting new class of materials and have attracted great attention because of their remarkable properties, such as high specific surface area, high porosity, high molecular alignment, and nano-size effects [118,119,120]. Additionally, many different types of molecules can be easily incorporated into nanofibers to greatly expand their drug-loading capacity or to provide a sustained release of the embedded drug molecules [121,122]. Nanofibers with continuous drug release properties that can enhance the efficacy of anticancer drugs, molecularly targeted drugs, and hyperthermia are being developed [116,123,124,125], and we believe that developments in this field will greatly benefit the sensitization of radiation.

### 3.5. Immunotherapy

The idea of using immunotherapy in combination with particle beam therapy is completely different from the combination of anticancer drugs and molecular-targeted drugs. It has been shown that local irradiation of tumors acts as an immune adjuvant and can elicit a systemic tumor immune response by killing tumor cells in situ [126,127,128]. Its basic mechanism is the induction of immunogenic cell death in the tumor microenvironment and the sequential activation of systemic cell-mediated immunity [129,130]. Danger signals and the release of tumor-specific antigens after ionizing radiation can turn the irradiated tumor into an in-situ vaccine. Recently, important angiogenic and immunosuppressive factors, such as vascular endothelial growth factor, interleukin 6 and 8, and hypoxia-inducing factor 1α, have been significantly downregulated after high-energy proton irradiation [131]. It has also been reported that proton irradiation mediates cell surface expression of calreticulin on tumor cells and increases susceptibility to cytotoxic T lymphocyte killing [132]. These findings suggest that PBT can inhibit angiogenesis and has an immunosuppressive mechanism, and thus, its therapeutic use can be expanded [133]. The C-ions correlated more with immune activation and prevention of metastasis in mouse models when used in combination with dendritic cell infusion [134,135,136]. In addition, in clinical cases, abscopal reactions have been reported in patients who received topical C-ion treatment [137,138].

In clinical studies using PBT, a phase I study in which an intratumoral injection of hydroxyapatite was used as an immune adjuvant after PBT was locally advanced or recurrent to prevent local or distant recurrence due to the activation of the immune system. It was performed in patients with hepatocellular carcinoma. The conditions of four of the nine patients were reported to have exacerbated [139]. Regarding C-ions, abscopal reactions have been reported in patients receiving topical C-ion treatment [137,138]. A promising approach to using this strategy to remove cancer cells that have spread to unirradiated areas is to combine particle therapy with immune system regulators, such as immune checkpoint inhibitors and cytokines. In addition, tumor-specific immune responses can be obtained by converting the tumor into an effective in situ vaccine using particle therapy.

## 4. Relationship between Boron Neutron Capture Therapy (BNCT) and Boron Compounds

### 4.1. The Principle of BNCT

The principle of boron neutron capture therapy was proposed four years after the discovery of neutrons by Chadwick in 1932 [140,141]. It uses ^10^B, which has a large absorbable cross section for thermal neutrons. It is one of the RT classified as particle beam RT and is characterized by the external irradiation of thermal neutrons, which have no electric charge, and the use of high-LET particle beams (alpha rays and lithium nuclei) produced by the reaction of neutrons with ^10^B in the body for cellular damage (treatment of malignant tumors). In principle, if boron can be directed to the target area, the reaction will occur only in that area and shows a cell-killing effect.

Briefly, it is common to use ^10^B as a compound that reacts easily with thermal neutrons, that is, a compound with a large absorbable cross section (3850 barns, 1 barn = 10^−24^ cm^2^). A ^10^B compound with tumor-accumulating properties is administered intravenously to the organism beforehand. When ^10^B has accumulated in the tumor, thermal neutrons are injected into the affected area by external irradiation. The neutrons react with ^10^B to produce alpha rays (helium nuclei) and lithium nuclei (Figure 4). Both of these have a short range of approximately 10 microns (about the size of a cell) and can provide high energy without exceeding the diameter of the cell. The principle is that the more selective the distribution of ^10^B to tumor cells, the less the damage to normal tissues and the higher the antitumor effect.

In RT, the accuracy of spatial positioning has been improved by increasing the accuracy of the treatment device and body sides. These techniques include respiratory synchronization, image-guided RT, and stereotactic RT. However, BNCT is a bimodal therapy, which is different from conventional RT in that it utilizes the difference in drug distribution, especially drug uptake between normal and tumor tissues, to achieve an effect. Thus, BNCT consists of two steps: the drug is responsible for tumor selectivity and the external irradiation of thermal neutrons is used to produce the effect. Therefore, it was necessary to develop a boron-containing compound with tumor-accumulating properties and a high-intensity thermal neutron source capable of delivering a sufficient dose (flux) for the treatment.

### 4.2. The History of BNCT

In the early days of research and development, the necessary thermal neutrons could only be obtained from experimental reactors. Clinical studies using experimental reactors were initiated in the United States in the 1960s, but the inadequate performance of ^10^B compounds prevented the discovery of promising results [142,143,144].

Since then, basic and clinical research has been conducted in Japan and Europe [145,146,147,148]. Neutron sources have generally been used through beam shaping, in which the beam from the core of a research reactor is attenuated to increase the proportion of low-energy thermal neutrons [149,150,151,152]. However, in recent years, as the use of nuclear reactors has become increasingly difficult for both commercial and research purposes, high-power neutron sources, such as the Japan Proton Accelerator Research Complex (J-PARK, Tokai, Japan), have been developed in the field of physics, and research on thermal neutron sources for therapeutic use without using nuclear reactors has become popular as a spin-out of these sources [153,154,155,156]. Accelerator neutron sources can be installed in hospitals for medical use, and they have a higher potential for medical applications than the use of nuclear reactors because they are not subject to the strict regulations of nuclear reactors.

Japan was the first country to begin its clinical application. An accelerator-based BNCT clinical study in Japan was developed based on the previous BNCT treatment in nuclear reactors, that is, malignant brain tumors and head and neck cancers were the first targets [157,158,159]. In June 2020, Stella Pharma’s L-*p*-boronophenylalanine (BPA) agents and Sumitomo Heavy Industries’ NeuCure were the first drugs and devices for BNCT to be covered by health insurance in Japan. As a result, BNCT has become a medical treatment option that utilizes drugs and medical devices, and its use as a full-fledged treatment for a limited number of indications has started. In addition, several research institutes and companies in the United States or China have succeeded in producing thermal neutron beams that can be installed in hospitals [160,161].

### 4.3. Reactor and Accelerator-Based Neutron Source

In terms of the medical use of nuclear reactors, the research reactor (JRR-4) at the Japan Atomic Energy Research Institute (the Japan Atomic Energy Agency) was modified for medical use in collaboration with the Department of Neurosurgery at the University of Tsukuba, with an operating room, simulation room, irradiation room, remote anesthesia equipment, and even experimental equipment [162,163]. With the development of a simulation calculation code, noncircumferential irradiation using epithermal neutrons can now be handled [164,165,166,167]. Epithermal neutrons are slightly higher in energy than thermal neutrons, and after entering the body, they become thermal neutrons and cause a neutron capture reaction, which has the advantage of allowing treatment of brain tumors with external irradiation [168,169]. While thermal neutrons themselves can only treat superficial lesions with a depth of approximately 2 cm, epithermal neutrons can treat lesions with a depth of approximately 6 cm, and the clinical application of epithermal neutrons will shift from intraoperative irradiation by craniotomy to external irradiation without anesthesia. The clinical application of thermal external neutrons has shifted from intraoperative irradiation by craniotomy to external irradiation without anesthesia.

In addition, unlike charged particles, neutron beams cannot be directly accelerated because they do not have an electric charge, and their handling is more difficult than charged particles [170].

### 4.4. Head and Neck Cancers

In head and neck cancer, the involvement of the human papilloma virus and the introduction of IMRT have reduced complications, and nivolumab has been introduced to treat recurrence. Head and neck cancer BNCT was used as a salvage treatment for patients who had already been irradiated and were at the time of recurrence or inoperable. It has attracted attention as a new treatment option for patients who have no other choice but palliative treatment, and because of the large number of cases compared to malignant brain tumors, it is thought to have been approved ahead of other treatments. The clinical study cases of head and neck cancer treated with BNCT are summarized in Table 4.

### 4.5. Malignant Brain Tumor

For malignant brain tumors, especially GBM, the combination of TMZ and radiation after maximum possible resection has been the standard of care since the breakthrough of TMZ [176], however, there has been no major technological innovation since then.

Malignant brain tumor BNCT has been used as the postoperative RT of choice in newly diagnosed cases or as salvage therapy in cases of recurrence [177]. The University of Tsukuba conducted a clinical study on newly diagnosed malignant glioma and showed that several protocols, including intraoperative irradiation of the craniotomy, prolonged survival by about two times compared to the standard treatment at that time using photon RT [178]. Chemotherapy at that time was in transition from nitrosoureas to TMZ. The median survival of 27 months was an epoch-making figure at that time, although the total number of patients was only a few dozen and there was selection bias, such as a lack of deep-seated disease. Similar to the head and neck region, clinical trials have been conducted for previously irradiated malignant gliomas that have recurred and are awaiting approval. Recent clinical studies on GBM treated using BNCT are summarized in Table 5. Malignant meningioma is another candidate for BNCT [179].

### 4.6. Requirements for Drugs for BNCT and Current Boron Compounds

Research and development of boron compounds for BNCT continues; however, the number of clinically available agents is very limited, with only two compounds, BPA and mercaptoundecahydro-closo-dodecaborate (BSH). The ideal BNCT drug would be able to accumulate ^10^B only in tumors and would not have any toxicity or effect on its own. The concentration required for the treatment depends on the intensity of the neutron beam.

BPA is the main component of borofalan, a newly approved drug for BNCT in Japan (Figure 5a). BPA is composed of the amino acid phenylalanine bound to a single atom of boron. The boron content is thus not high (approximately 5% by weight). It is believed to be absorbed into the cytoplasm via amino acid transporters in tumor cells with an active metabolism. Since tyrosine is a substrate of melanin, it was first used in BNCT for malignant melanoma [146], and its accumulation effect was later found in many histological types of tumors; it was also used for head and neck cancer and malignant brain tumors [185,186]. BSH is a low molecular weight compound with a molecular weight of approximately 200 and has been used for malignant brain tumors (Figure 5b) [187,188]. They do not accumulate in tumors and penetrate tissues by diffusion, and they do not cross the blood-brain barrier in normal brain tissues. This point was reversed in a BNCT clinical study using BSH for malignant brain tumors, which utilized the fact that malignant brain tumors normally diffuse into the interstitium only around disrupted tumor blood vessels. Inevitably, there are also studies using protocols that expect additive effects by combining multiple drugs, that is, BPA and BSH in a two-drug combination, but at present, BSH alone is not used because of the complexity of the procedure and the low tumor/blood ratio.

In addition, as a technological development for diagnosis, fluorine-labeled BPA (^18^FBPA) has been developed for positron emission tomography (PET) examination [189,190,191]. The advantage of being able to determine the accumulation of boron drugs before BNCT treatment has been found to reliably predict the therapeutic effect of BNCT.

BNCT is a single-irRT, and the thermal neutron fluence (n/cm^2^: the number of particles passing through a unit area) required for a single treatment is approximately 1 × 10^13^ (within one hour of treatment), which can be generated by current accelerators. To achieve tumor control with this neutron fluence dose, a tissue or intracellular ^10^B concentration of 20–40 μg/mL is required. To maintain this concentration for the duration of irradiation, the dose of drug administered to the body must be extremely high and must be administered for a short period, and for BPA, a dose of 500 mg/kg is required. According to the dosage and administration approved in Japan, 30 g of the compound was administered in 3 h to a human weighing 60 kg. Compared to anticancer drugs and antibiotics, where the dosage ranges from a few hundred milligrams to a few grams at most, 30 g is a huge amount. Of course, the boron drug alone does not have an antitumor effect, but it does need to accumulate in the tumor.

The ideal conditions for a boron compound for BNCT are:The concentration of ^10^B in the tumor tissue or cells must be at least 20 μg/mL during neutron irradiation.It must be safely administered and excreted.No toxicity is observed in bolus doses of several tens of grams.It must be water-solubleThe tumor/normal tissue (T/N) ratio or tumor/blood (T/B) ratio should be as high as possible. The result is a drug with high therapeutic efficacy and reduced damage to normal tissues.

Attempts to develop new boron compounds that are more effective and safer have been ongoing for a long time, and applications of nanoparticles using drug delivery systems, antibody drugs, low-molecular-weight boron compounds have been studied and attempts to improve the tumor/normal tissue ratio as a standard have been ongoing for a long time.

### 4.7. Development of New Boron Compounds

#### 4.7.1. Problems with Existing Boron Compounds and New Drug Development

While general low-molecular-weight anticancer drugs show cell-killing effects at extremely low concentrations of 10^−6^ to 10^−9^ M in tumors, the concentration of boron agents that show therapeutic effects in BNCT is extremely high, at 10^−3^ M in the tumor (Figure 5c). Therefore, according to the guidelines for the development of BNCT irradiation systems, it is desirable to have a T/B ratio and a T/N ratio of ≥3.0, from the viewpoint of safety [192]. However, BPA, which is currently widely used in BNCT, is rapidly excreted from the tumor by transporters, and its retention in the tumor is low. In May 2020, steboronin was launched as the world’s first boron drug for BNCT in Japan, but its dosage is extremely high at 9000 mg/300 mL. In addition, to achieve excellent therapeutic effects in BNCT, it is important not only to maintain high concentrations of boron drugs in the tumor but also to distribute them homogeneously. Masunaga et al. have shown that quiescent cancer cells have reduced BPA uptake compared with proliferating cancer cells [193,194]. Since it has been reported that quiescent cells include hypoxic cells and CSCs that are resistant to various cancer therapies, including radiation [195,196,197], it can be inferred that the heterogeneous uptake distribution greatly affects the therapeutic effect of BPA-BNCT. Therefore, it is highly desirable to develop a delivery technology that enables tumor-selective and efficient delivery of boron drugs in BNCT therapy.

#### 4.7.2. Drug Development Using Existing Boron Compounds

As a method for the highly efficient delivery of boron drugs, investigations using liposomes [198], antibody modification [199], and dendrimers [200] have been reported. Kueffer et al. reported that tumor accumulation could be improved by encapsulating BSH in liposomes and improving blood retention, but since BSH also accumulated in normal cells, a strategy to add tumor cell selectivity is necessary [201] Kang et al. reported that BSH was efficiently taken up by cancer cells by encapsulating it in liposomes modified with a peptide that has a high affinity for integrin αvβ3, which is highly expressed in neovascularization and GBM [202]. Maruyama et al. found that boron drugs encapsulated in PEGylated liposomes improved tumor retention and maintained tumor BPA levels at 30 μg/g tissue for at least 72 h after intravenous administration [203]. Furthermore, Nomoto et al. reported that BPA gel formed by mixing polyvinyl alcohol (PVA) and BPA could inhibit the excretion of BPA from tumors by allowing BPA to enter cells through endocytosis [204]. We also tried to improve the active accumulation of one of the existing boron compounds, BSH, to cancer using ND201, which is cyclodextrin modified with folic acid (Figure 6) [205]. As a result, ND201 induced active cancer accumulation and high retention of BSH depending on the expression level of folic acid receptors in cancer cells and succeeded in obtaining excellent antitumor effects using neutron irradiation.

#### 4.7.3. Development of Next-Generation Boron Drug for Fusion with Drug Delivery System Technology

As mentioned above, a drug delivery system (DDS) that can control the intracellular uptake pathway and intracellular retention must be developed for BNCT. There are two concepts of DDS in solid tumor therapy: active targeting and passive targeting. Active targeting utilizes the specific binding ability of molecules for targeting. Passive targeting utilizes the characteristics of the tumor vascular system to achieve selective tumor accumulation. In general, macromolecules do not easily leak from normal blood vessels. However, in solid tumors, macromolecules tend to leak from tumor blood vessels because of characteristics, such as increased neovascularization and vascular permeability. In addition, because of the immaturity of the lymphatic system and other reasons, macromolecules that leak locally in the tumor remain there for a long time. As a result, high-molecular-weight anticancer drugs that are highly stable in the blood can be passively targeted [206,207]. This is the enhanced permeability and retention (EPR) effect. The EPR effect, first announced in 1986, has been accepted worldwide and has contributed to the development of methods to deliver anticancer drugs, nucleic acids, genes, and peptides to cancer tissues in polymeric polymers, liposome preparations, and micelle preparations at the animal experimental level.

Nakamura et al. focused on the structure of lipids that form liposomes and developed boron lipids, distearoyl boron lipid (DSBL), and boron cholesterol with chlossodecaborate [208], and confirmed the long-term survival after thermal neutron irradiation in a mouse colon cancer CT26 cell transplantation model with an intratumor boron concentration of more than 170 ppm [209]. Nishiyama et al. synthesized PEG-b-P (Glu-SSH), a biodegradable poly(ethylene glycol)-poly(glutamic acid) block copolymer containing BSH disulfide bonded to mercaptoethylamine, and developed nanomicelles by self-assembly [210]. After administration of 50 mg B/kg to mice implanted with CT26 mouse colon tumor, the intratumor boron concentration reached 69 ppm after 24 h, indicating a T/B concentration ratio of more than 20 and a strong BNCT antitumor effect. In addition, Nakamura et al. developed a boron compound, maleimide-functionalized closo-dodecaborate (MID), which can be introduced into cysteine [211], and focused on the phenomenon that stained serum albumin accumulates in tumor tissue (EPR effect), maleimide-functionalized closo-dodecaborate albumin conjugates (MID-AC) was prepared by binding MID to serum albumin and verified the antitumor effect of BNCT using cancer transplant mice [212]. Nagasaki et al. developed nanomicelles (PM micelles) in which vinylcarborane was polymerized in biodegradable polyethylene glycol-polylactic acid block copolymer micelles [213]. In an experiment using tumor-bearing mice, PM micelles reached 14 ppm after 24 h of administration, and the tumors disappeared in two out of five mice after 25 days of thermal neutron irradiation. More recently, a collaboration between our group and the Nagasaki laboratory has led to the development of PBA-modified polymeric nanoparticles (Nano^PBA^) with sialic acid orientation as a boron preparation that has different targeting properties from LAT1 and can maintain ^10^B levels in tumors for a long time [214]. Nano^PBA^ has a supramolecular structure consisting of a core and shell composed of poly(lactic acid) (PLA) and poly(ethylene glycol) (PEG) segments (Figure 7). PBA is located at the hydrophilic end of the polymer, and a large number of PBAs are exposed on the surface of the nanoparticles and bind to the membranes of cancer cells in multiple ways, resulting in a very strong targeting effect. In this study, the efficacy and safety of Nano^PBA^ were verified by administering Nano^PBA^ to a B16 melanoma-bearing mouse model that highly expresses sialic acid, followed by irradiation with thermal neutron radiation. The results showed that Nano^PBA^ showed a longer intratumor accumulation time than BPA and showed a potent antitumor effect by efficient tumor targeting even at a low dose of 1/100. Furthermore, focusing on the anaerobic glycolytic system, which is characteristic of cancer, a collaboration between our group and Maeda laboratory has developed a multifunctional boron compound, styrene-maleic acid (SMA)-glucosamine borate complex, [215]. The complex is a novel boron compound with multiple functions that can induce changes in the metabolic system and suppress the growth of cancer by the contained boric acid and glucosamine and can also induce cell lethality by BNCT (Figure 8). A remarkable cancer growth inhibitory effect was confirmed by irradiating a cancer-bearing mouse model transplanted with Chinese hamster squamous cell SCCVII with thermal neutrons by administering 125 mg/kg of SMA-glucosamine borate complex. This was an antitumor effect equivalent to that of the 500 mg/kg BPA-administered group, and the SMA-glucosamine borate complex was able to show efficacy at ^10^B, which is 1/70 of that of BPA.

#### 4.7.4. Challenges in Conducting BNCT Research

At present, there are no clinical studies using these new compounds, except for phase II studies conducted on boronated porphyrin. In Japan, recurrent head and neck cancers are covered by public insurance, and the number of BNCT treatment facilities is expected to increase, as well as the number of applicable diseases and patients. As a result, the number of researchers involved in the development of new boron agents is expected to increase.

Further focus on the development of new drugs is desirable, with the understanding that the doses of boron compounds are orders of magnitude higher than those of conventional drugs, and that they do not need to have their drug effects, and therefore require tumor accumulation and more stringent safety evaluation. In other words, as mentioned above, it is clear that the development of new boron preparations is urgently needed in addition to the radiobiological research of BNCT to further expand the indications for BNCT. However, the number of neutron sources that can be used for BNCT is limited, and there are only a few facilities in the world where biological and chemical experiments using cells and especially laboratory animals are possible. We at the University of Tsukuba have been developing a new Linac accelerator BNCT device for several years and have already completed a device that can withstand the practical level (Figure 9). The facility also has a biological laboratory where basic research on cells and small animals (mouse and rat) irradiated with an accelerator BNCT device is possible, and animal experiments can be performed with a GLP-compliant grade. In the near future, we plan to publish a large number of research results on new boron drug candidates using this facility.

## 5. Concluding Remarks

The main purpose of combination therapy using existing therapies such as chemotherapy, molecular targeted agents, and radiation is to control the local area using radiation and to suppress metastasis, including micrometastasis, using chemotherapy and molecular targeted agents. In recent years, this trend has been changing, and combination protocols aimed at synergistically increasing the effect on the local area have been considered but have not been established. Owing to the limited number of facilities for particle therapy, there have been few results from both basic and clinical applications, and in principle, it has been considered to follow the conventional combination method in photon RT. However, with the recent remarkable development in the field of nanoparticles and biomaterials, the possibility of proposing new drugs and new combination therapies that better match the characteristics of particle RT is being demonstrated. In this review, we summarized the status of combination therapies in clinical practice with a focus on particle RT and summarized the latest findings on various combination therapies that are being clarified from basic biology, to provide an opportunity to consider combination therapies that should contribute to the next generation of particle RT for cancer.

The current particle therapy has achieved sufficient progress in the field of physical engineering, such as the miniaturization of accelerators and freedom of irradiation direction using gantries. However, the biological benefits are not yet fully understood, leaving the potential for clinical contributions. One of the reasons for this is that the uncertainty of biological problems is much greater than that of physical problems. Therefore, from a clinical point of view, in addition to the biological effects of particle beams, it is highly desirable to clarify the advantages and limitations of using particle beams in combination with other biological therapies. The expected benefits of combined RT include: (1) radiosensitization of tumor tissue (ideally tumor-specific), (2) protection of normal tissue, and (3) induction of bystander or abscopal effects in distant regions.

In PBT, combination therapy has been attempted based on the experience of photon RT, and the combination of radiation and cytotoxic chemotherapy has become the standard treatment for most locally advanced cancers. Because of its excellent dose distribution and mild biological effects, PBT can reduce the exposure dose of normal tissues, which may have side effects when used in combination with other therapies (Cox 2007). In contrast, the physical and biological properties of carbon ions are very different from those of photons, and while they can produce unexpected results, as in the immunotherapy described above, the risks are undeniably greater than expected.

Finally, BNCT, which is now covered by health insurance in Japan, is a different type of RT that must be used in combination with drugs. In recent years, researchers in the fields of drug development and drug delivery have become interested in radiation cancer therapy, probably because of the development and awareness of BNCT. In addition to the accelerator-based BNCT, our facility is equipped with experimental facilities for physics, chemistry, biology, and medicine, and we promise to make a significant contribution to future BNCT research. We hope that drug research in the field of BNCT will have a significant impact on the development of combination therapies for other types of particle therapies.

## Figures and Tables

**Figure 1 jpm-11-00825-f001:**
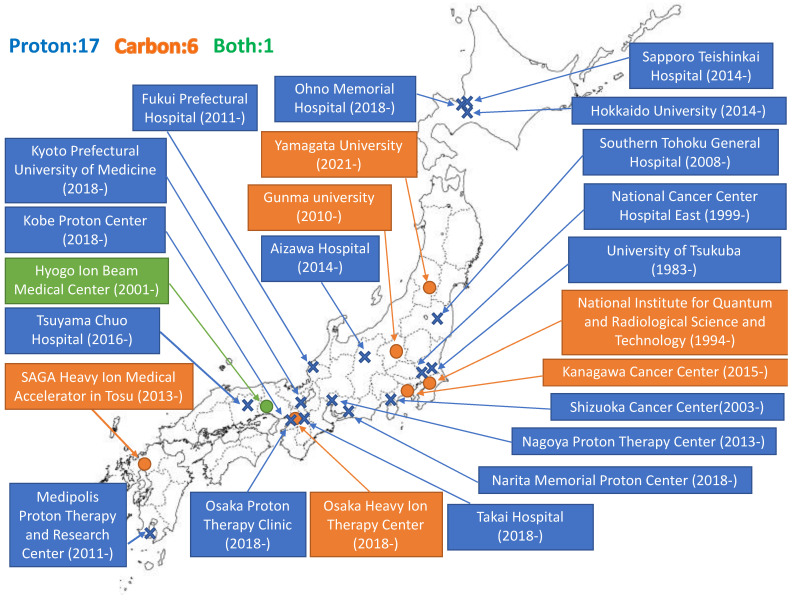
Expansion of particle beam therapy facility in Japan.

**Figure 2 jpm-11-00825-f002:**
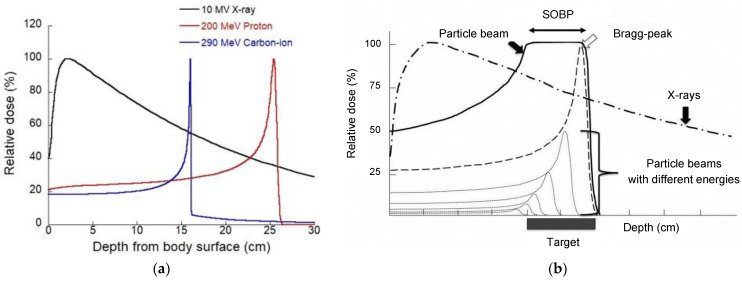
(**a**) Depth-dose distributions of clinical X-ray, proton, and carbon-ion beams and (**b**) the spread-out Bragg peak (SOBP).

**Figure 3 jpm-11-00825-f003:**
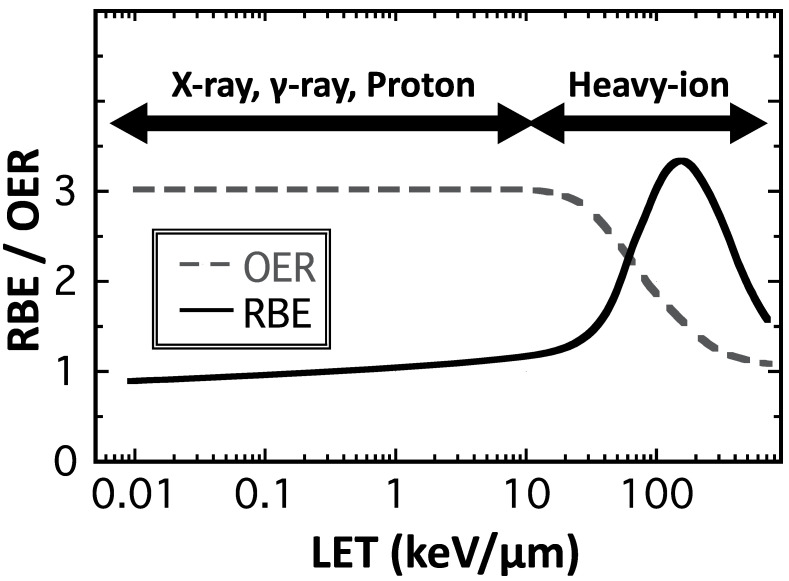
Relationship between RBE, OER, and LETs.

**Figure 4 jpm-11-00825-f004:**
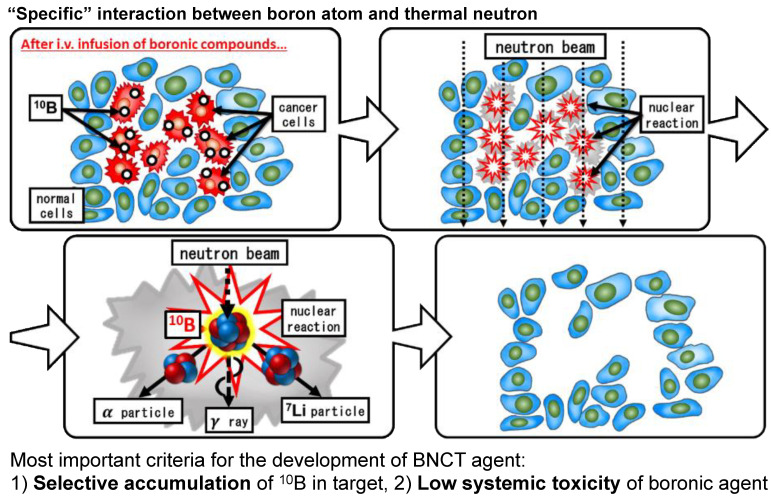
Selective cell destruction using boron neutron capture therapy (BNCT).

**Figure 5 jpm-11-00825-f005:**
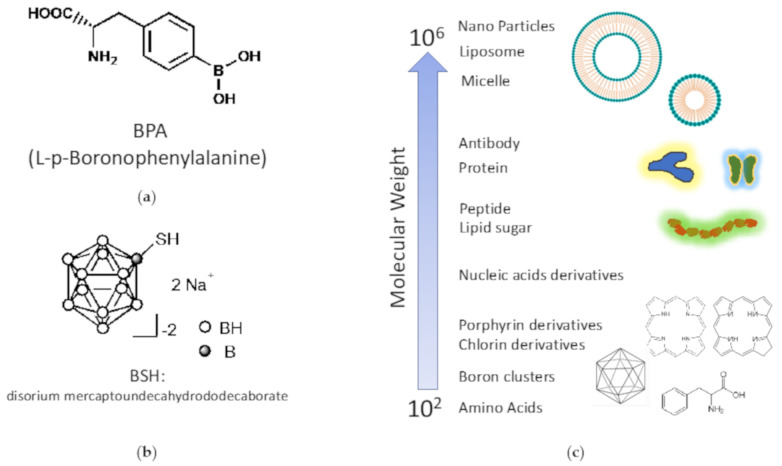
Chemical structures of (**a**) BPA and (**b**) BSH, and (**c**) molecular weight of various compounds.

**Figure 6 jpm-11-00825-f006:**
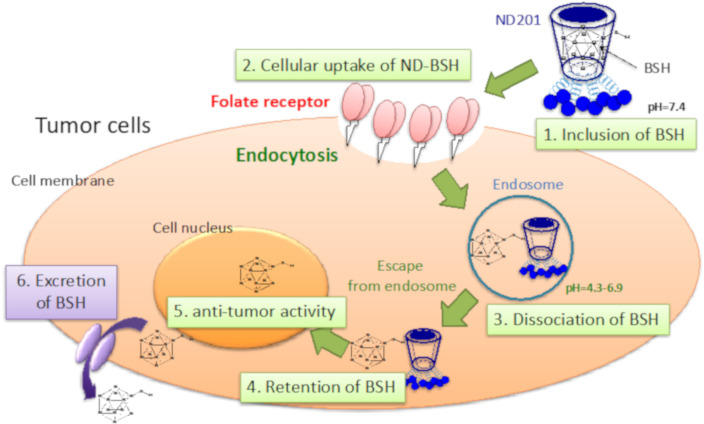
The mechanism of action of ND-BSH. BSH is included in the lumen of ND201 and recognizes cancer cells targeting the folate receptor.

**Figure 7 jpm-11-00825-f007:**
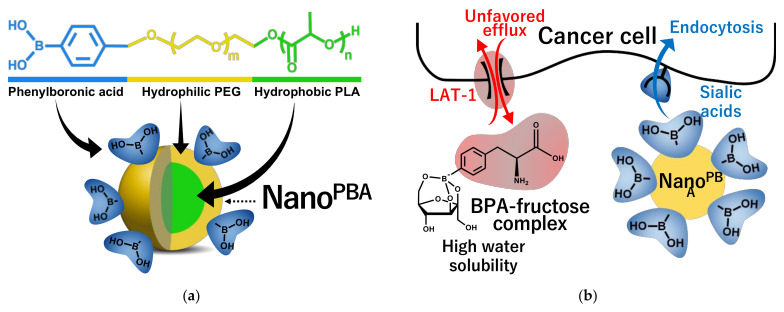
(**a**) Schematic illustration of molecular design for PBA-decorated polymeric nanoparticle as a novel BNCT agent and (**b**) different accumulation mechanism from BPA-f.

**Figure 8 jpm-11-00825-f008:**
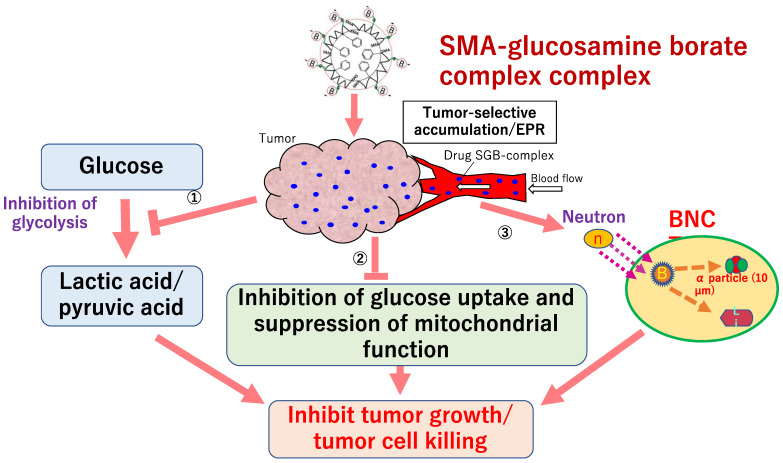
Multifunctional cancer growth inhibitory effect by SMA-glucosamine borate complex targeting treatment-resistant cancer cells.

**Figure 9 jpm-11-00825-f009:**
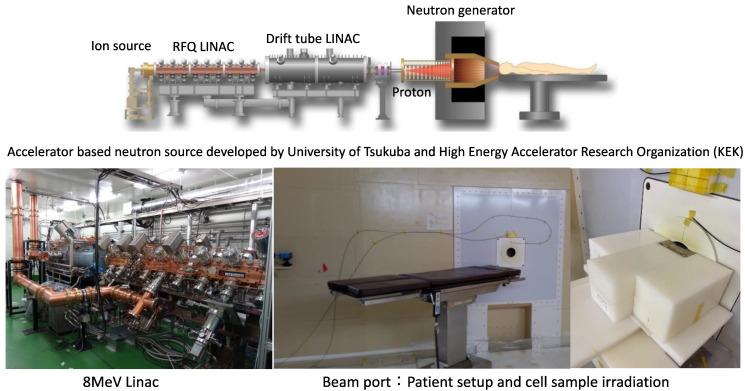
Newly developed accelerator-based neutron source (Tsukuba model).

**Table 1 jpm-11-00825-t001:** Cardiopulmonary toxicities of photon RT and PBT for esophageal cancer.

Author	N	RT Modality	Treatment	Endpoint	Late Toxicity Rate	
					Heat	Lung
DeCesaris [15]	36	Photon RT	Preope/definitive	Perioperative death	13.9%	
	18	Proton			0%	
Wang [6]	320	IMRT	Preope/definitive	Grade 3 (2y/5y)	18%/21%	NA
	159	Proton			11%/13%	NA
Wang [17]	208	3DCRT	Preoperative	Perioperative complication	15.9%	30.3%
	164	IMRT			17.1%	23.8%
	72	Proton			9.7%	13.9%
Makishima [10]	19	3DCRT	Definitive	Grade 3	0%	10.3%
	25	Proton			0%	0%
Xi [18]	211	IMRT	Preope/definitive	Grade 3	2.4%	4.7%
	132	Proton			0.8%	2.3%
Lin [19]	61	IMRT	Preope/definitive	Grade 3	5 *	11 *
	46	Proton			3 *	5 *

RT, radiotherapy; IMRT, intensity-modulated radiotherapy; 3DCRT, three-dimensional conformal radiotherapy; NA, not assessed; *, number of events.

**Table 2 jpm-11-00825-t002:** Concurrent particle beam therapy combined with chemotherapy for pancreatic cancer.

Author	N	RT	Dose	Chemotherapy	Treatment	Outcome
Hong [46]	25	proton	30GyRBE/10fr 25GyRBE/5fr	capecitabine	preoperative	OS: 75%/1Y
Terashima [48]	50	proton	50GyRBE/25fr 70.2GyRBE/26fr 67.5GyRBE/25fr	gemcitabine	curative	OS: 76.8%/1Y, PFS: 64.3%/1Y
Hong [47]	50	proton	25GyRBE/5fr	capecitabine	preoperative	OS: 42%/2Y
Maemura [52]	10	proton	50GyRBE/25fr 67.5GyRBE/25fr	gemcitabine, S-1	curative	OS: 80, 45, 22.5%/1, 2, 3Y
Kim [49]	37	proton	45GyRBE/10fr	capecitabine, 5-FU	curative	OS: 75.7%/1Y, PFS: 64.8%/1Y, 19.3M
Jethea [50]	13	proton	50GyRBE/25fr	capecitabine, 5-FU	curative	OS: 62, 40%/1, 2Y, 16M
Hiroshima [51]	42	proton	50–67.5GyRBE/25-33fr	gemcitabine, S-1	curative	OS: 77.5, 50.8%/1, 2Y, 25.6M
Kawashiro [53]	72	carbon	52.8GyRBE/12fr 55.2GyRBE/12fr	gemcitabine, S-1 (n = 56)	curative	OS: 73, 46%/1, 2Y, 21.5M
Vitolo [58]		carbon	38.4GyRBE/4fr	FOLFIRINOX, gemcitabine	preoperative	

Value in outcome represents overall survival rate, progression-free survival rate, and median survival time. OS: overall survival, PFS: progression-free survival.

**Table 3 jpm-11-00825-t003:** Clinical outcomes of photon and PBT trials for prostate cancer.

Author	N	RT Modality	Total Dose (Gy)	Photon (Gy)	Proton (GyRBE)	Efficacy (%)	Late toxicity (Grade 3) (%)	
							GI	GU
Shipley [67]	202	Photon +Proton	75.6 67.2	50.4 (pelvis) 50.4 (pelvis) + 16.8 (local)	25.2 (local)	8y-LC:73	2.9	NA
		Photon			-	59	0	NA
Roach [68]	440	Photon	70.2 70.2	50.4 (pelvis) + 19.8 (local) 70.2 (local)		7y-PFS:40	4.3	3
		Photon				27	0	0
Local prostate irradiation								
Zeitman [69]	393	Photon +Proton	79.2 70.2	50.4 (local) 50.4 (local)	28.8 (local)	10y-bRF:83	1	2
		Photon +Proton			19.8 (local)	67	0	2
Kuban [70]	301	Photon	78.0 70.0	78.0 (local) 78.0 (local)		10y-FFF:73	7	3
		Photon				50	1	5

GI, gastrointestinal; GU, genitourinary; LC, local control; PFS, progression-free survival; bRF, biochemical relapse-free; FFF, freedom from any failures.

**Table 4 jpm-11-00825-t004:** The summary of the clinical study cases of head and neck cancer treated using BNCT.

Facility	Neutron Source	Year	Tumor	Patients No.	Boron Agents	Clinical Course
Osaka University [171,172]	KUR JRR4	2001–2014	Rec H&N	45	BSH, BPA	5y 32% 10y 21% PR 29% CR 51%
Kawasaki Medical College [173]	KUR JRR4	2003–2011	Rec H&N	20	BPA	PR 35% CR 55%
Kawasaki Medical College	KUR JRR4	2006–2012	H&N preop.	7	BPA	5y 42% PR 1/7 CR 5/7
Helsinki University Central Hospital [174]	FiR-1	2003–2008	Rec H&N	30	BPA	MST 13mo PR31% CR 45%
Taipei Veterans General Hospital [175]	THOR	2010–2011	Rec H&N	10	BPA	PR 40% CR 30%

KUR: Kyoto Research Reactor, Kumatori, Osaka, Japan; JRR4: Japan Research Reactor No.4; Tokai, Ibaraki, Japan; FiR-1: Finland Reactor 1, Otaniemi, Finland, THOR: Tsing Hua open-pool Reactor, Hsinchu Taiwan, Rec H&N: recurrent head and neck cancer, H&N preoperative: head and neck cancer patients of preoperative state, BSH: disodium mercaptoundecahydrododecaborate, BPA: L-*p* boronophenylalanine, 5y: 5 year survival rate, 10y: 10 year survival rate, PR: Partial response rate, CR: Complete response rate, MST: Median overall survival time.

**Table 5 jpm-11-00825-t005:** The summary of the recent clinical study cases of glioblastoma treated using BNCT.

Facility	Neutron Source	Year	Tumor	Patients No.	Boron Agents	Clinical Course (Month)
University of Tsukuba [178]	JRR4	1998–2007	GBM	15	BPA, BSH	MST 23.3 27.1
Tokusima University [180,181]	KUR JRR4	1998–2008	GBM	23	BSH	MST 15.5 19.5 26.2
Osaka Medical College [182,183,184]	KUR	2002–2006	GBM	21	BPA, BSH	MST 14.5 23.5
		2002–2007	rGBM	19	BSH, BPA	MST 10.8

KUR: Kyoto Research Reactor, Kumatori, Osaka, Japan; JRR4: Japan Research Reactor No.4, Tokai, Ibaraki, Japan; GBM: glioblastoma multiforme; rGBM: recurrent glioblastoma; BSH: disodium mercaptoundecahydrododecaborate; BPA L-*p* boronophenylalanine; MST, median overall survival time.

## Data Availability

The data that support the findings in this article are available from the corresponding author, Y.M., upon reasonable request.

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
