# Peer review of "A Critical Review of Radiation Therapy: From Particle Beam Therapy (Proton, Carbon, and BNCT) to Beyond"

_jpm, 2021, doi:10.3390/jpm11080825_

Round 1

Reviewer 1 Report

In this paper,  the authors discussed the clinical and basic aspects of particles-based radiation therapy (RT)  mainly, proton beam therapy (PBT), carbon-ion beam therapy (CIBT), and boron neutron capture therapy (BNCT).  The review was well prepared and I strongly recommend for possible publication of the manuscript with the present form.

Author Response

Thank you for your supportive comment. Following your comments, our manuscript has been reviewed in English by a native speaker. As a result, the title has also been partially changed.

Reviewer 2 Report

I had the opportunity of reviewing this manuscript that privides a thorough review and discussion of the role of particle therapy in multidisciplinary treatment. 

This paper is generally of interest and provides the community with insights related to the use of Particle Beam Therapy, its current situation and future clinical  expectations. The authors have disected the three particle beam therapy modalities and the final product represent an excellent review.  

The only inor suggestion would be to incorporate the existing experience wtith the mentioned radiation techniques in pediatric oncology 

Author Response

Thank you for your supportive comment. Following your comment, I also added the status of proton therapy for childhood cancer.

p8 Line5-p9 Line14 (2.4. Pediatric cancer)
